



# Linking Woody Plants, Climate, and Evapotranspiration in a Temperate Savanna

Horia G. Olariu[1], Bradford. P. Wilcox[1], Sorin C. Popescu[1]

[1]Ecology and Conservation Biology, Texas A&M University, College Station, TX, 77843, USA

*Correspondence to*: Horia G. Olariu (horia@tamu.edu)

**Abstract.**

Evapotranspiration is the dominant pathway by which water returns from land surfaces and vegetation to the atmosphere in many semiarid and subhumid regions. In this study, we integrated satellite-based estimates of evapotranspiration with climate, runoff, and woody-vegetation data to evaluate how changes in precipitation, temperature, and canopy cover jointly influence

water loss in a temperate savanna that spans both semiarid and subhumid climates. Our validation at the sub-basin scale showed that modeled evapotranspiration agreed moderately well with water-balance estimates (coefficient of determination ≈ 0.65, bias −7 millimeters per water year, and root mean square error 103 millimeters per water year). Across the region, annual evapotranspiration totals generally reached about 90 percent of precipitation, indicating an ecosystem strongly driven by atmospheric water demand. In dry years, water loss occasionally exceeded rainfall, highlighting a heightened sensitivity to soil

moisture shortages and extreme heat. Areas with high woody-canopy cover consistently exhibited higher evapotranspiration and lower net water surplus. Notably, where canopy cover exceeded 80 percent in the driest portions of the study area, the soil water surplus turned negative over multiple years. These findings underscore the potential for expanding woody cover to limit groundwater recharge and reduce overall water availability, especially under warming and more variable precipitation regimes. Future work could explore fine-scale, long-term impacts of woody plant density and targeted management strategies that

optimize trade-offs among vegetation growth, ecosystem health, and water resources.

## 1 Introduction

Evapotranspiration (ET) is the principal flux returning water from the Earth's surface to the atmosphere, with estimates suggesting that 62,000–75,000 km³ of water cycles back annually (Abbott et al., 2019). This process accounts for more than 60% of global precipitation (Oki & Kanae, 2006). Because the difference between precipitation (P) and ET is commonly used

to evaluate water availability at catchment and basin scales (Falkenmark et al., 1989), accurately quantifying ET and identifying its key drivers are critical for effective water resource management and ecosystem protection.

Climatic factors—temperature (T) and precipitation—are typically regarded as the main drivers of ET (Dai et al., 2018). Rising temperatures have increased atmospheric moisture demand worldwide, manifesting as upward trends in potential evapotranspiration (PET)—the theoretical maximum ET assuming no water limitation (Feng & Fu, 2013; Fu et al., 2016;



Scheff & Frierson; Zhao & Dai, 2015; Zhao & Dai, 2016). Unlike PET, which ignores water constraints, measured ET is closely coupled with P, generally displaying a positive correlation (Stocker et al., 2013). However, the strength of this coupling varies across regions, climates, and timescales. In contrast, T and ET exhibit a weaker relationship overall. Although they are more strongly correlated in humid areas, they may decouple and even show negative correlation under arid conditions during extreme heat events (Yuan et al., 2020; Alessi et al., 2022; Qiu et al., 2020; Berg & Sheffield, 2018).

In addition to climatic factors, ecosystem structure—particularly changes in woody vegetation—can significantly alter ET rates. Numerous studies in the United States document how woody plant encroachment (WPE), defined as the expansion of native trees and shrubs into grass-dominated systems such as grasslands and open-canopy savannas (Acharya et al., 2018), modifies ET. In Texas, Dugas et al. (1998) and Afinowicz et al. (2005) observed ET decreases of 31.9 mm and 110 mm, respectively, following the removal of *Juniperus ashei*. Dugas et al. (1998) further noted that these decreases persisted

only for two years, after which the effect diminished. By contrast, in Oklahoma, Wang et al. (2018) reported a 45% increase in mean annual ET in a former grassland region after its conversion to *Juniperus* spp.–dominated woodlands; and Qiao et al. (2015) showed that average ET rates in *Juniperus virginiana* woodlands were 100 mm/yr higher than those in neighboring grasslands. Similar patterns appear farther west, in a riparian area in Arizona, where *Prosopis velutina* woodlands exhibited an ET rate of 692 mm/yr, compared with 548 mm/yr for an adjacent grassland (Scott et al., 2014).

The Post Oak Savannah ecoregion of east-central Texas presents a particularly compelling set of conditions for a case study examining how climate and woody vegetation jointly influence ET. Over the past 150 years, anthropogenic reshaping of this landscape has resulted in a mosaic of grasslands, savannas, and densely wooded thickets (Campbell, 1925; Tharp, 1926; McBride, 1933; Parmalee, 1955; Garza & Blackburn, 1985; Midwood et al., 1998; Singhurst et al., 2004; Griffith et al., 2007; Stambaugh et al., 2011). Recent remote sensing studies by Olariu et al. (2024) revealed that between 1996 and 2022, ca. 9.7%

(5,338 km²) of the Post Oak Savannah underwent WPE, converting grassland and open-canopy savanna into woodland, while another ca. 6.8% (4,504 km²) experienced "thicketization," marked by proliferating sub-canopy woody plants in established woodlands. At the same time, some 5.7% showed the opposite trend, transitioning from woodlands to more open savanna or grasslands. Superimposed on these rapid land-cover changes are pronounced east–west gradients in precipitation (850–1250 mm/yr) and temperature (18–22°C), with drier, hotter conditions in the southwest and cooler, wetter conditions in the northeast

(Schmidly, 2002). These dynamic biophysical conditions underscore the importance of studying how changing vegetation structure and climate interact to shape ET across this region.

        Shifts in ecology and biodiversity associated with thicketization in oak savanna systems have been thoroughly examined (Brudvig & Mabry, 2008; Brudvig & Evans, 2006; Zirbel et al., 2017). However, the hydrological implications of WPE in these water-limited ecosystems remain comparatively understudied. Because ET is generally the dominant component

of the water budget (Condon et al., 2020; Reitz et al., 2017; Seager et al., 2018), an increase in woody cover could substantially alter water cycling. Indeed, a recent study in the Post Oak Savannah by Basant et al. (2023) found that thicketization markedly reduced deep drainage and, in some cases, halted groundwater recharge altogether. Meanwhile, woodlands that had not undergone thicketization still experienced recharge, but at much lower rates than non-thicketized areas. Although these





findings strongly suggest that ET increases in response to woody plant proliferation, this hypothesis remains unquantified—creating a clear knowledge gap regarding how WPE affects water resources in oak savanna ecosystems.

To address this knowledge gap, the present study integrates remote sensing and hydrological modelling approaches to characterize ET dynamics across the Post Oak Savannah between 2008 and 2023. We employ MOD16A2GF C6.1, hereafter referred to as MOD16—the gap-filled, eight-day net ET dataset—along with water-balance estimates to validate and refine ET measurements at multiple temporal scales. By combining these satellite-derived products with spatially explicit woody plant metrics and climate data, we aim to determine how variations in vegetation structure and environmental conditions influence ET. Accordingly, this study pursues four primary objectives: (1) Validate MOD16 in the Post Oak Savannah by comparing satellite-derived ET data against water-balance estimates, thereby establishing the accuracy of MOD16 for regional-scale analyses; (2) Examine monthly and seasonal variations in ET, gaining insight into short-term and interannual changes; (3) Analyze the relationship between woody plant metrics (canopy cover and canopy height), climatic factors (precipitation and temperature), and ET at the water-year scale to quantify how shifts in vegetation composition and climatic drivers affect ET rates, and (4) Evaluate evapotranspiration–precipitation ratios (ET/P) and calculate excess water (precipitation minus evapotranspiration, P – ET) across the region at the water-year scale to provide a broader assessment of water availability under varying woody cover and climatic conditions. By integrating a robust remote sensing framework with field-based validation and detailed ecological data, this study aims to enhance our understanding of how climate and WPE jointly influence water cycling in the Post Oak Savannah.

## 2 Materials and Methods

### 2.1 Study Site

The Post Oak Savannah ecoregion in east-central Texas covers over 55,000 km² and spans 31 counties, with its western boundary encompassing much of the Carrizo–Wilcox Aquifer (Fig. 1A). Historically, this region supported an open-canopy savanna characterized by diverse grasses and forbs interspersed with stands of post oak (*Quercus stellata*) and blackjack oak (*Quercus marilandica*) (Wasowski & Wasowski, 1988). Positioned between the East Texas Piney Woods—dominated by dense evergreen forests—and the Central Texas Blackland Prairie—characterized by black, calcareous, alkaline, clay-rich soils—this landscape functions as an ecological transition zone (Diggs et al., 1999; Schmidly, 2002).

During the study period (2008–2023), both precipitation and temperature displayed pronounced spatial variability, with annual precipitation ranging from approximately 1,400 mm in the northeastern portion of the ecoregion to about 600 mm in the southwest (Fig. 1B). Mean annual temperature exhibited a similar gradient, decreasing from roughly 22°C in the northeast to 17°C in the southwest (Fig. 1D). Canopy cover over this interval showed substantial fragmentation, with densely wooded stands interspersed among open, grass-dominated areas (Fig. 1E). In contrast, the southern portion exhibited a more continuous mosaic of cover types, with less abrupt transitions between wooded and non-wooded patches (Fig. 1E).





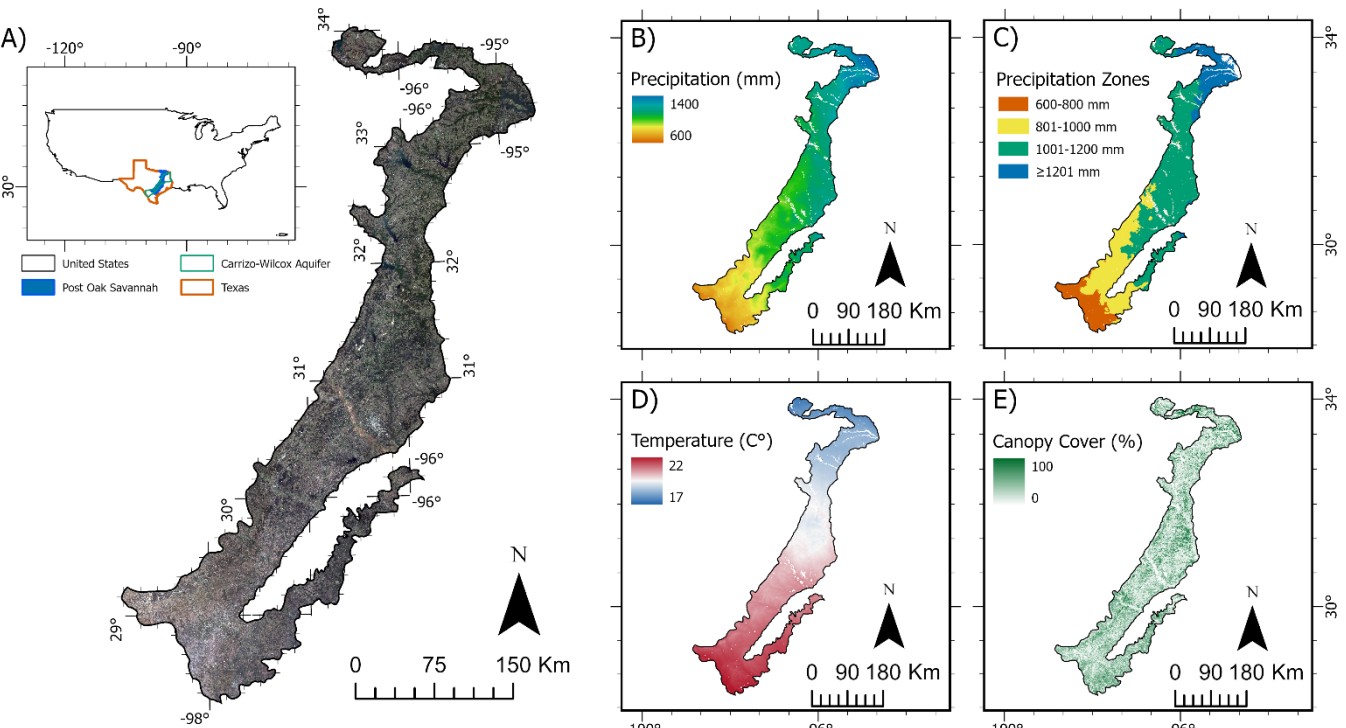

**Figure 1.** Overview of the Post Oak Savannah ecoregion in east-central Texas. The left side of Panel (A) shows the ecoregion's location relative to the United States, Texas, and the Carrizo–Wilcox Aquifer; the right side provides a true-color Landsat 8 satellite mosaic of the Post Oak Savannah (2023). Panels (B) and (C) illustrate, respectively, mean annual precipitation (mm) and a classification of the four precipitation zones over the period 2008–2023. And panels (D) and (E) depict, respectively, temperature (°C) and canopy cover for the same timeframe.

## 2.2 Data and Preprocessing

### 2.2.1 Evapotranspiration Data

In this study we used the MOD16 Collection 6.1 terrestrial ecosystem ET dataset, which is based on a modified Penman–Monteith formulation (Mu et al., 2007; Mu et al., 2011). It provides 8-day cumulative ET estimates for the global land surface at a 500-m spatial resolution (units: mm/m2). This ET product has been widely employed in global ET modeling (Kim et al., 2012; Ershadi et al., 2015; Ramoelo et al., 2014; Trambauer et al., 2014; Velpuri et al., 2013), showing variable performance upon validation but generally stronger accuracy in North America (Velpuri et al., 2013; Zhang et al., 2019). Developed using meteorological data from NASA's Global Modeling and Assimilation Office (GMAO) and various MODIS-based inputs (e.g., LAI, FPAR, albedo) (Mu et al., 2011), Version 6.1 offers notable improvements in areas affected by clouds and/or aerosol contamination. Specifically, it employs a year-end gap-filling technique in which 8-day intervals lacking reliable FPAR/LAI data are replaced with the average of the best available FPAR/LAI for that vegetation pixel over the preceding five years.

A quality control workflow was implemented to exclude bad pixels from the analysis. Pixels produced solely by the MOD16 backup algorithm were masked and removed. Additionally, since MODIS employs its own confidence quality score



assessment, only pixels with scores of 0 and 1—indicating good and usable data—were retained, while all others were
discarded. Finally, we used the MCD12Q1.061 MODIS Land Cover Type Yearly Global 500-m Land Cover Type 1: Annual
IGBP classification system to mask pixels classified as Water Bodies, Barren, Cropland, or Cropland/Natural Vegetation
Mosaic. These were excluded from the analysis because of the lack of natural vegetation and the influence of artificial watering
on the results.

### 2.2.2 Temperature and Precipitation Data

The temperature and precipitation products used in this study were obtained from the Daymet V4 model, developed by the
Oak Ridge National Laboratory and supported by NASA through the Earth Science Data and Information System (Thornton
et al., 2022). Daymet provides long-term, continuous, gridded estimates of daily climate variables at a 1-km resolution by
interpolating and extrapolating ground-based observations via statistical modeling techniques. It has been widely utilized in
ecological, hydrological, and agricultural studies (Akinsanola et al., 2024; Dey et al., 2024; Bhat et al., 2024; Zahura et al.,
2024; Bennemann et al., 2023). Because Daymet provides daily minimum and maximum temperatures, we calculated the
simple mean for each day to derive the average daily temperature.

### 2.2.3 Woody Plant Metric Data

Two primary metrics were used to characterize woody vegetation in this study: canopy cover and canopy height. The canopy
cover data originated from Version 3 of the Rangeland Analysis Platform (RAP), developed by the University of Montana in
partnership with the U.S. Department of Agriculture (USDA). This dataset combines tree and shrub cover to capture the full
spectrum of woody plants influencing ET (Allred et al., 2021). The RAP cover estimates integrate information from 75,000
field plots and the historical Landsat record. Through cloud computing and temporal convolutional networks, annual
predictions are generated at a 30-m resolution across the United States. Validation against approximately 7,500 field plots
yielded mean absolute errors (MAE) of ±6.2% and ±2.6% for shrubs and trees, respectively, and root mean square errors
(RMSE) of ±8.8% and ±6.7% for shrubs and trees, respectively. While RAP has primarily been applied in agricultural contexts
(Hudson et al., 2021; Morford et al., 2022; Subhashree et al., 2023; Retallack et al., 2023), it also has demonstrated utility in
ecological studies (Olariu et al., 2024).

Canopy height data were drawn from two sources: Potapov et al. (2021), which provides 2019 estimates, and
Malambo and Popescu (2024), which supplies 2020 estimates. Potapov et al. (2021) produced a 30-m-canopy height model
(CHM) by extrapolating canopy height measurements from Global Ecosystem Dynamics Investigation (GEDI) footprints to
analysis-ready Landsat data, using a bagged regression tree ensemble method (Breiman, 2001). When validated against
airborne lidar, the CHM displayed an RMSE of 9.07 m, an MAE of 6.36 m, and an $R^2$ of 0.61, performing particularly well
for taller trees (≥10 m). This dataset has largely been employed to quantify stocking rates and biomass for ecological research
(Ali & Rahman, 2025; Dröge et al., 2025; Potapov et al., 2022; Hawker et al., 2022). In contrast, Malambo and Popescu (2024)
integrated ICESat-2 (Ice, Cloud, and Land Elevation Satellite-2) with ancillary Landsat, LANDFIRE, and topographic



variables to produce a 30-m-canopy-height product. Validation against airborne lidar (R² = 0.72, MAE = 3.9 m) revealed higher accuracy in densely forested environments—such as mangroves, coniferous forests, or mixed broadleaf forests—than in sparsely vegetated regions like deserts and chaparral. Although relatively new, this product has already been applied to hurricane-impact studies in mangrove ecosystems (Roy et al., 2024) and other remote sensing research (Guo et al., 2024; Guenther et al., 2024).

### 2.2.4 Runoff Data

The runoff data used for the water balance ET (WBET) calculations were obtained from USGS WaterWatch (http://waterwatch.usgs.gov), a platform that provides streamgage-based maps for over 3,000 long-term (30 years or more) USGS streamgages. Runoff was calculated at the water-year scale for each HUC8 subbasin by dividing the average daily flow for the water year by the drainage basin area, and it was assumed to be uniform across the entire basin.

### 2.2.5 Stacking and Aggregation

All projection, resampling, and aggregation for this study were performed on the Google Earth Engine (GEE) platform (Gorelick et al., 2021). To align the various datasets, each was projected to the EPSG:3857 (Spherical/Web Mercator) coordinate system, clipped to the Post Oak Savannah boundary (U.S. EPA Level 3 ecoregion), and resampled using the 500-m MOD16 grid (Omernik & Griffith, 2014).

To maintain consistency with the ET product, both canopy-cover and canopy-height datasets were resampled from 30 m to 500 m via mean resampling, which preserved the continuous nature of the data (Blan & Butler, 1999). By contrast, Daymet data were resampled from 1-km to 500-m using the nearest-neighbor method to retain the original values (Brandsma & Können, 2006).

Once aligned and resampled, the datasets were aggregated to monthly scales, water-year scales (October 1 to September 30), and overall averages for the entire study period. For instance, Water 2009 encompasses data from October 1, 2008 through September 30, 2009. This water-year approach was chosen in lieu of the standard calendar year (January 1 to December 31) to better capture the lagged effects of the region's precipitation patterns—rainier fall and spring seasons and drier summers—on vegetation and water balance (Null & Viers, 2013; He et al., 2021; Papacharalampous & Tyralis, 2020). Specifically, the 8-day, 500-m MOD16 ET product was aggregated to monthly (January 2008–December 2023) and water-year (2009–2023) scales. Pixels that had been masked during any portion of a particular month or water year were given a null value and excluded from analysis. The daily, 500-m Daymet V4 precipitation dataset was aggregated to both monthly and water-year intervals, whereas temperature was aggregated only to the water-year scale. Precipitation data were further averaged across the entire study period and then grouped into 200-mm precipitation zones (600–800 mm, 801–1000 mm, 1001–1200 mm, and ≥1201 mm) to assess the influence of varying aridity (Figure 4-1C). To align the annual canopy cover and canopy height metrics with the ET data, each year's canopy values were matched to the corresponding water year's ET (e.g., canopy cover for 2012 was compared with ET from Water Year 2012), ensuring that nine of the twelve months overlapped. Lastly,



the canopy cover dataset was also aggregated into an overall average spanning the study period for use in monthly analyses and the excess water analysis.

Finally, to enhance our understanding of water use and cycling in areas of increasing woody vegetation density, canopy cover was stratified into six classes: 0–10%, 11–20%, 21–40%, 41–60%, 61–80%, and ≥81%. These distinctions were informed by an extensive literature review encompassing a wide range of ecological and hydrological considerations. Numerous studies identify 10% canopy cover as the upper threshold for grasslands in temperate climates (Dixon et al., 2014; Plappert et al., 2024; Hu, 2024). By contrast, savanna systems typically exhibit between 10% and 60% canopy cover

(Loewensteiner et al., 2021; Anchang et al., 2020), with higher percentages generally characterizing tropical savannas, where woodlands are denser than in temperate zones. Accordingly, the 10%–60% range was subdivided into three strata: 10%–20% cover, representing transitional grassland–open-canopy savannas; 21%–40% cover, representing open-canopy savannas; and 41%–60% cover, representing savanna–woodland transition zones. The 41%–60% range is more prevalent in the northern Post Oak Savannah, where higher precipitation supports greater woody density. The highest cover categories (61%–80% cover and

≥81%) were then designated as woodlands and thicketized woodlands, respectively.

## 2.3 Major Steps

Consistent with the four objectives of this study, we (1) validate the MOD16 ET product against water-balance estimates (WBET) at the subbasin (HUC8) scale; (2) analyze monthly and seasonal ET differences as they relate to canopy cover; (3) use linear regression to examine the coupling and decoupling of woody plant metrics (canopy cover and canopy height),

climatic factors (precipitation and temperature), and ET within different precipitation zones; and (4) evaluate excess water (P – ET) at the water-year scale—including an ET/P analysis—to assess broader trends in water availability over the study period (Fig. 2).




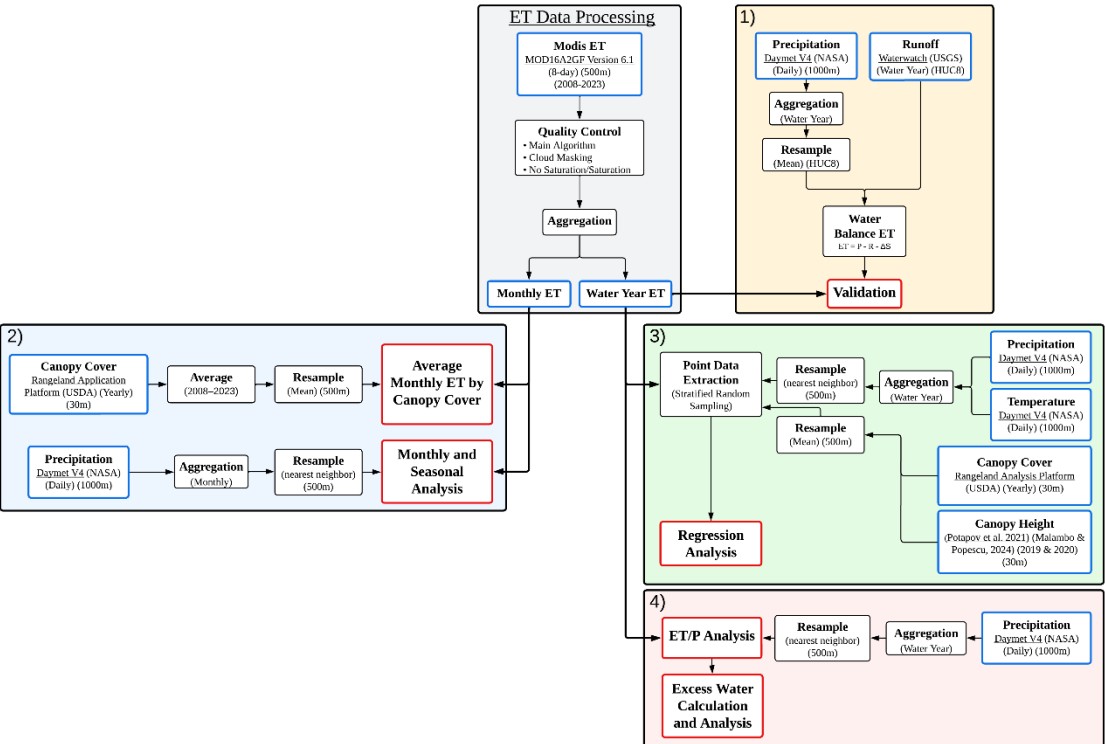

**Figure 2.** Overview of the data processing and analysis workflow used in this study. The grey panel outlines how monthly and water-year
ET from the MOD16 product is derived via quality control, cloud masking, and aggregation. The yellow panel (1) illustrates validation of
the MODIS-based ET product by deriving water balance ET (WBET = P – R – ΔS) from precipitation (Daymet V4) and runoff (USGS
WaterWatch) at the subbasin scale. The blue panel (2) shows the analysis of monthly and seasonal ET in relation to canopy cover from the
Rangeland Analysis Platform and precipitation from Daymet V4. The green panel (3) shows the application of regression analyses on
precipitation, temperature, canopy cover, and canopy height to assess how woody plant metrics and climate factors influence ET across
different precipitation zones. And the pink panel (4) illustrates the evaluation of overall water availability by comparing ET with precipitation
(ET/P) and calculating excess water (P – ET) at the water-year scale.

### 2.3.1 MOD16 ET Validation

At the water-year scale, WBET for HUC8 subbasins was compared with MOD16 ET. The water-year WBET for these HUC8
sub-basins was computed as follows:

$$1) \quad WBET = P - R - \Delta S \quad,$$

where P, R, and ΔS are water-year precipitation, runoff, and storage changes at HUC8 subbasins, respectively.

The independent WBET dataset we used to compare against the MOD16 ET estimates, was generated via a water
balance approach at the HUC8 scale. The conterminous United States is partitioned into hierarchical hydrologic units, each
assigned a unique hydrologic unit code (HUC) consisting of two to eight digits (Seaber et al., 1987). The largest unit is a region
(HUC2), followed by a sub-region (HUC4), a basin (HUC6), and ultimately a subbasin (HUC8).

Following established methods in the literature, we applied several filters to exclude HUC8 subbasins where the water
balance was unlikely to close (i.e., WBET ≠ P – R). First, we removed any HUC8s having a runoff-to-precipitation ratio (R/P)



exceeding 0.40, to mitigate the influence of regional groundwater flow (Velpuri et al., 2013; Senay et al., 2016). We also excluded HUC8s having a WBET greater than PET and those having less than 60% of their area located within the Post Oak Savannah. These criteria resulted in 11 HUC8s being retained (Fig. 3). Among them, the percentage of area within the Post Oak Savannah ranged from 61 to 99%, with an average of 76%. In total, 154 pairwise comparisons (11 HUC8s × 14 water years) were available, because WaterWatch data extended only to the 2022 water year.

All 154 paired points were plotted and the $R^2$, Bias, and RMSE were calculated. Furthermore, $R^2$, Bias, and RMSE were calculated for each HUC8, as well as each water year.

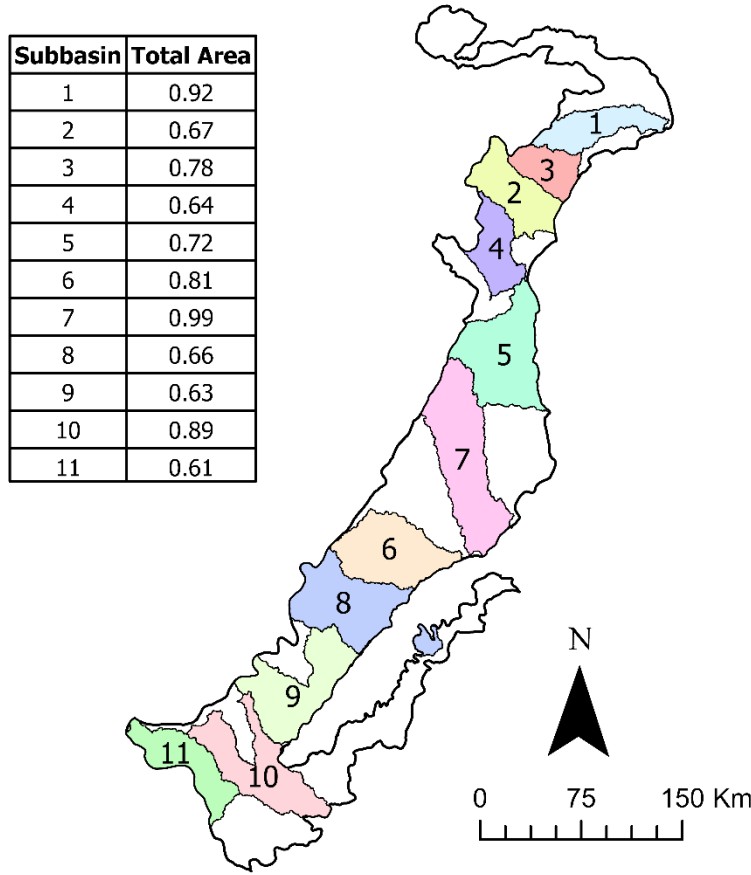

| Subbasin | Total Area |
|---|---|
| 1 | 0.92 |
| 2 | 0.67 |
| 3 | 0.78 |
| 4 | 0.64 |
| 5 | 0.72 |
| 6 | 0.81 |
| 7 | 0.99 |
| 8 | 0.66 |
| 9 | 0.63 |
| 10 | 0.89 |
| 11 | 0.61 |

**Figure 3.** Map of the 11 retained HUC8 subbasins within the Post Oak Savannah, color-coded by subbasin ID. The table lists the percentage of each subbasin area contained within the ecoregion.

### 2.3.2 Monthly and Seasonal Analysis

Monthly MOD16 ET was averaged across the entire study period (2008–2023) to obtain monthly mean values. These monthly means were then extracted for each cover class and precipitation zone. Finally, the 12 monthly means for each class and zone were summed to calculate annual averages and standard deviations.





### 2.3.3 Point Data Extraction and regression Analysis

The 500-m water-year products, spanning 15 water years, were compiled for each variable, with ET designated as the response variable and Precipitation, Temperature, Canopy Cover, and Canopy Height serving as predictors. Next, a random stratified sampling approach was implemented to extract 1,000 points per precipitation zone, yielding a total of 4,000 points containing ET, Precipitation, Temperature, Canopy Cover, and Canopy Height for each water year. Points with missing values for any product in any water year were excluded from further analysis, resulting in 3,550 points for the regression models.

Each predictor was then paired with its corresponding ET value from the same water year (e.g., 2009 Canopy Cover with 2009 ET) and plotted. Simple linear regressions were conducted to generate lines of best fit and determine R² for each predictor–response pair, within each precipitation zone. This approach facilitated an examination of how the relationships between these variables vary under different levels of long-term aridity.

### 2.3.4 ET/P and Excess Water Analysis

Over the entire study period, total ET and P values were aggregated across the Post Oak Savannah. The ratio of ET to P (ET/P) was then computed to facilitate further analysis of the fraction of precipitation lost to the atmosphere. Next, the total ET was subtracted from the total P to quantify the volume of excess water retained in the terrestrial system. Finally, these excess water values were averaged within each cover class and precipitation zone.

## 3 Results

### 3.1 MOD16 ET Validation

Comparisons of the MOD16 product with WBET estimates yielded an $R^2$ of 0.65, a bias of -7 mm wyr$^{-1}$ (−0.8%), and an RMSE of 103 mm wyr$^{-1}$ (11.6%) (Fig. 4). Among individual HUC8s, $R^2$ ranged from 0.11 to 0.70, bias spanned −79 to 85 mm wyr$^{-1}$, and RMSE varied between 63 and 104 mm wyr$^{-1}$. Examined by water year, $R^2$ ranged from 0.04 to 0.80, bias extended from −125 to 117 mm wyr$^{-1}$, and RMSE ranged from 57 to 127 mm wyr$^{-1}$ (Fig 4).





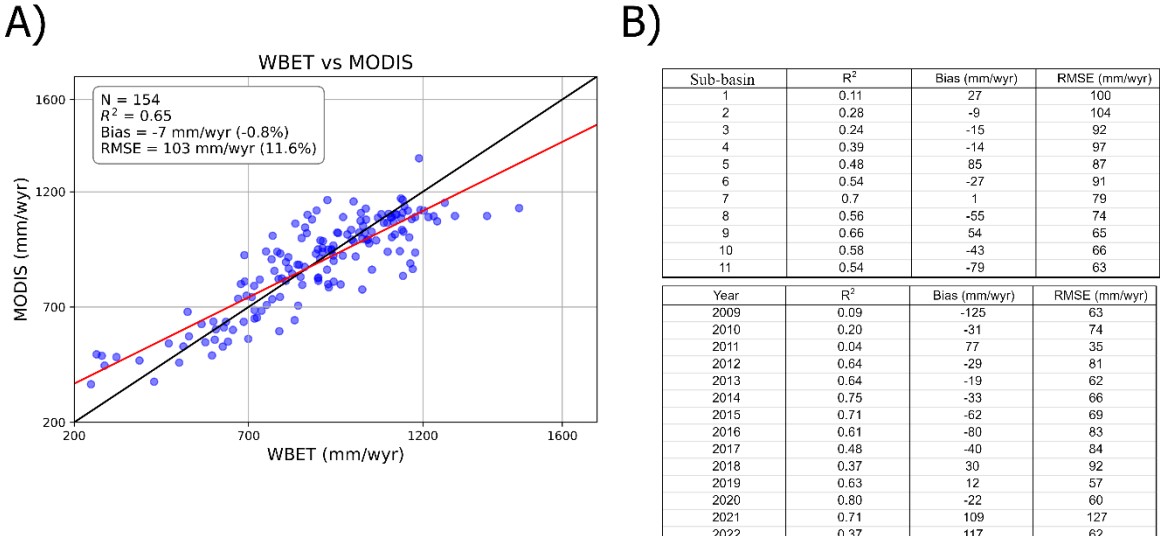

**Figure 4.** Comparison of Water-Balance Evapotranspiration (WBET) and MOD16 ET at the HUC8 subbasin and annual scales (2009–
255 2022). (A) Scatterplot of WBET (x-axis) versus MODIS-estimated ET (y-axis); the solid black line represents the 1:1 line and the red line
is the linear regression fit. The inset box summarizes sample size (N), coefficient of determination ($R^2$), bias (mm wyr$^{-1}$ and %), and root
mean square error (RMSE in mm wyr$^{-1}$ and %). (B) Tables showing $R^2$, bias, and RMSE for each HUC8 sub-basin (top) and each water year
(bottom)

### 3.2 Monthly and Seasonal Analysis

Monthly ET increases from January to June, peaking at 133.6 mm (averaged across all canopy classes), before dropping to
32.2 mm in December (Fig. 5). The highest single ET value, 168.0 mm, occurs in June within the ≥81% canopy cover class,
while the lowest single value, 30.5 mm, is observed in December within the 0%–10% cover class. The ≥81% cover class
exhibits the highest ET values for six months (April–September), whereas the 61%–80% cover class dominates in the
remaining months (Fig. 5).



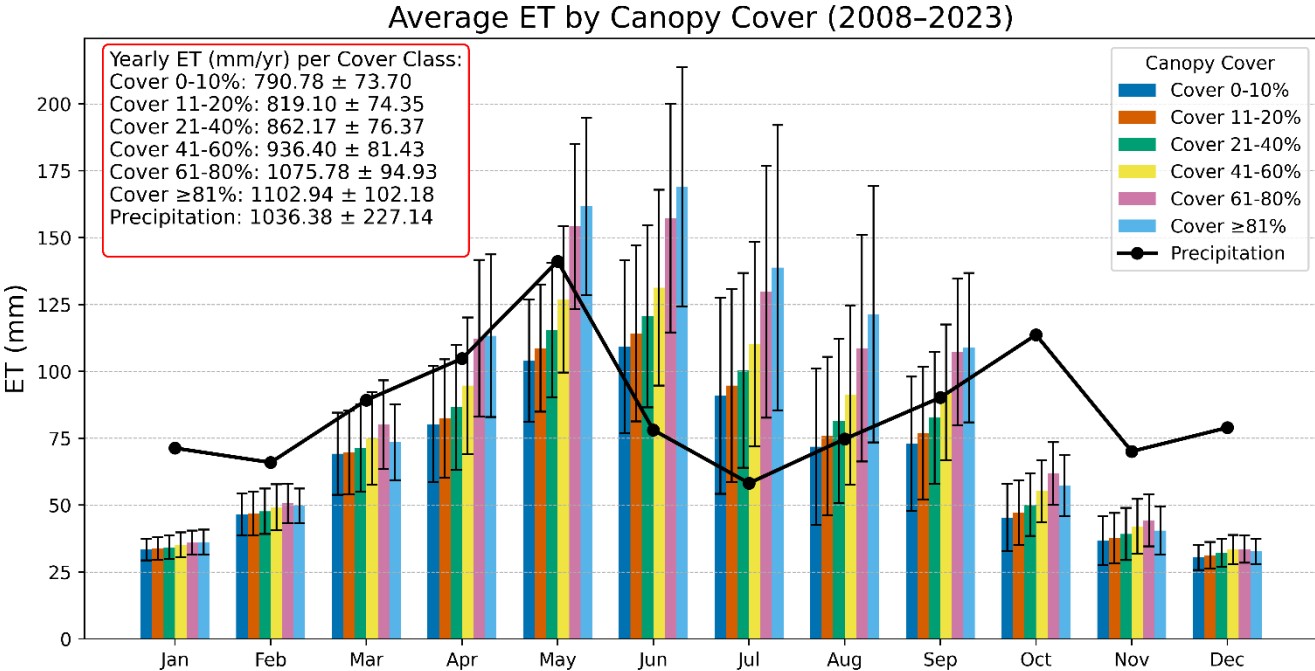

**Figure 5.** Monthly average ET by canopy-cover class (and corresponding precipitation (2008–2023).

ET exhibited a seasonal cycle, with average lows of 38.5 mm in winter (December–February), moderate levels of 98.7 mm in spring (March–May), and peak levels of 112.1 mm in summer (June–August), before declining to 61.0 mm in fall (September–November) (Fig. 6). Notably, the drought year 2011 showed substantially lower ET values relative to other years—averaging 78.7 mm in spring, 42.6 mm in summer, and 36.1 mm in fall—coinciding with the low precipitation totals. Conversely, higher precipitation levels led to greater distinction between seasonal ET averages (Fig. 6).






**Figure 6.** Time-series of monthly ET and precipitation from 2008 to 2023. The top panel shows monthly ET (black line) alongside seasonal average lines. The bottom panel displays monthly precipitation (black bars), the two horizontal lines indicating overall average monthly ET for 2010–2014 and 2015–2021.

### 3.3 Regression Analyses

The two climatic variables P and T exhibited notably different relationships with ET. Precipitation showed a moderate positive correlation, with $R^2$ values ranging from 0.23 (1001–1200 mm) to 0.61 (600–800 mm) and slopes from 0.29 (1001–1200 mm) to 0.55 (≥1201 mm) (Fig. 7). By contrast, temperature demonstrated a weak negative relationship with ET, with $R^2$ values varying from 0.09 (801–1000 mm) to 0.27 (600–800 mm) and slopes between −82.02 (600–800 mm) and −45.72 (801–1000 mm) (Fig. 7).





In comparison, the two woody-vegetation metrics showed more consistent positive relationships with ET than the climatic variables. Canopy height exhibited a moderate positive correlation, with R² values between 0.48 (600–800 mm and ≥1201 mm) and 0.54 (801–1000 mm), and slopes ranging from 35.05 (1001–1200 mm) to 38.66 (801–1000 mm) (Fig. 7).

Similarly, canopy cover displayed a weaker but still positive association, with R² values ranging from 0.12 (600–800 mm) to 0.20 (1001–1200 mm) and slopes between 3.61 (600–800 mm) and 4.32 (1001–1200 mm) (Fig. 7).

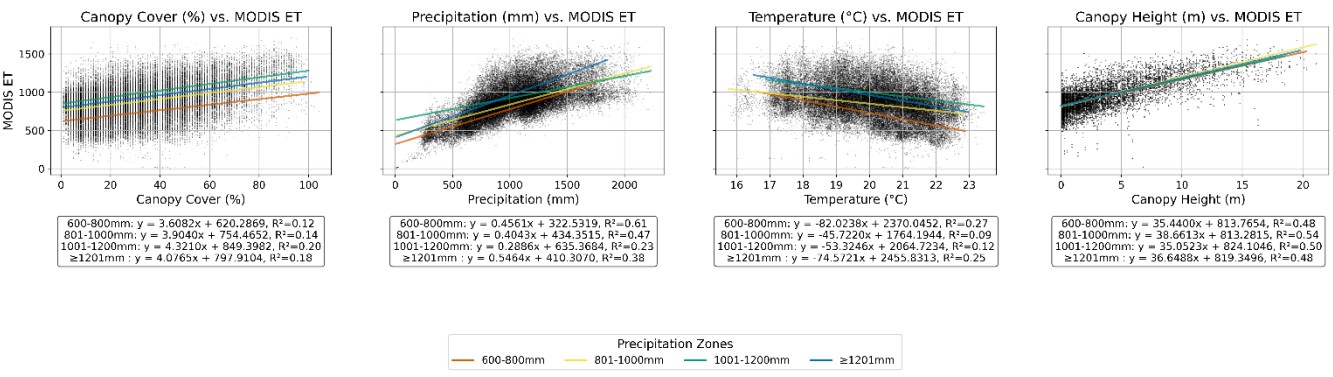

**Figure 7.** Scatterplots illustrating the linear relationships between MODIS ET (y-axis) and four predictors—precipitation, temperature, canopy cover and canopy height (x-axes)—for different precipitation zones (600–800 mm, 801–1000 mm, 1001–1200 mm, ≥1201 mm). The
four precipitation zones are represented by color-coded lines showing the best fit regression for each, and the corresponding slope, intercept, and R² values are shown in the insets. Canopy height data were available only for 2019 and 2020.

### 3.4 ET/P Ratios and Excess Water Analysis

The ratio of ET to P (ET/P) remained relatively stable, ranging between 70% and 100% throughout the study period, with an overall mean of 90% (Fig. 8). Notable deviations occurred in 2011 and 2022, when ET/P exceeded 100%. Both years were
characterized by above-average temperatures and below-average precipitation (Fig. 8).





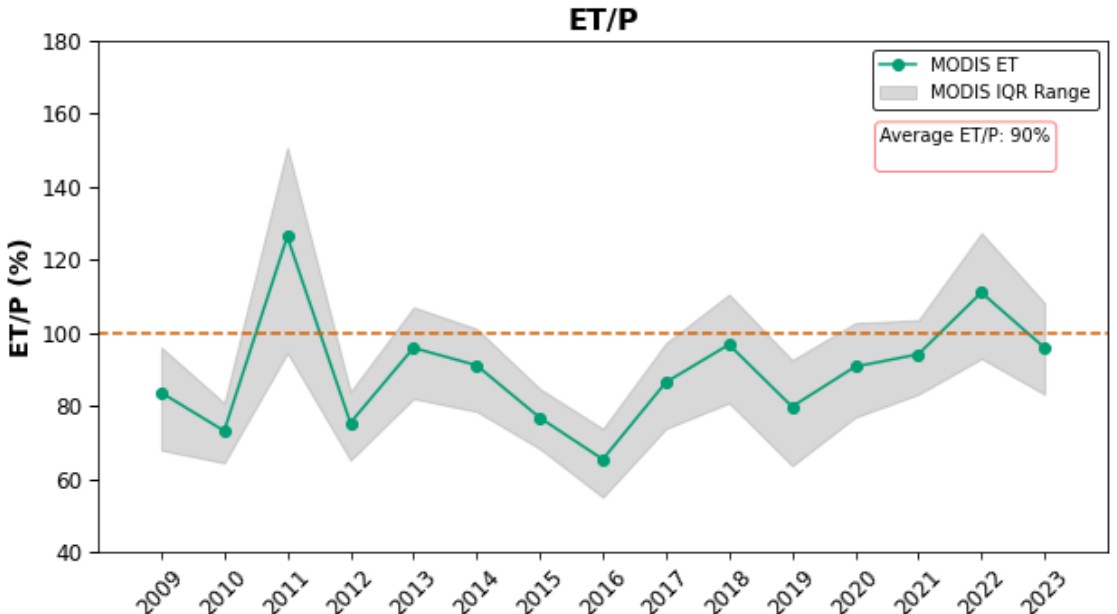

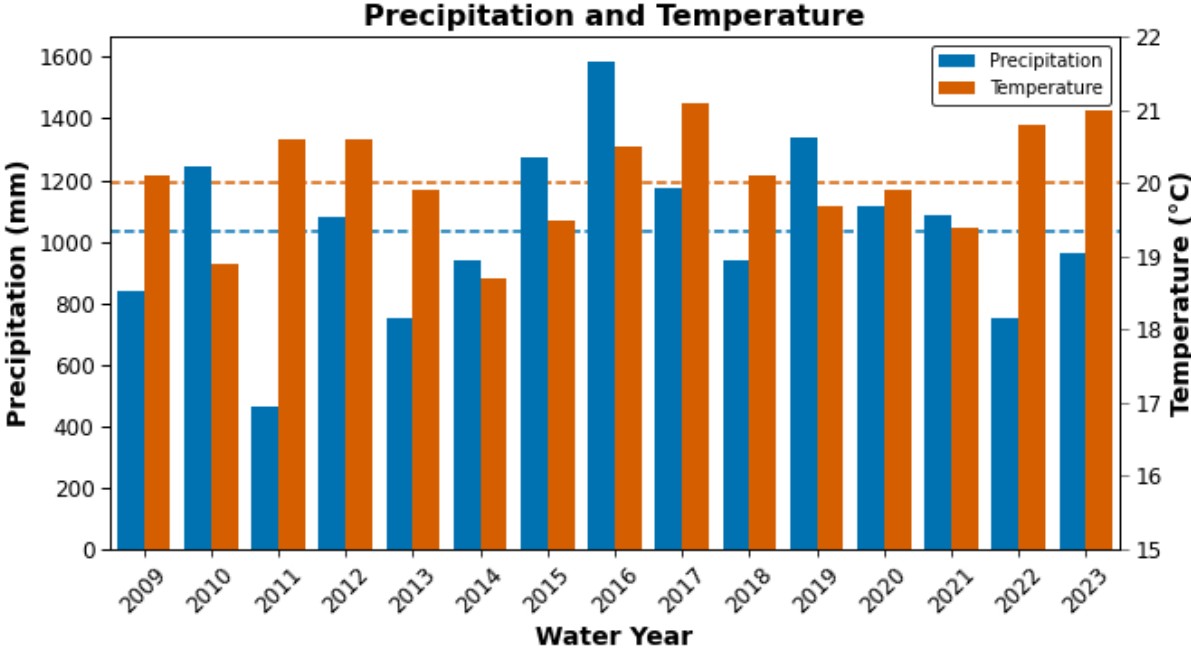

**Figure 8.** (Top panel) Annual ET/P ratios derived from MODIS ET estimates, with the interquartile range (IQR) shown in gray and the overall mean ratio (90%) by the dashed red line. (Bottom panel) Corresponding water-year precipitation and temperature with dashed lines denoting average precipitation and temperature over the study period.






Areas exhibiting ET/P ratios above 100% in the Post Oak Savannah predominantly coincided with low-elevation, forested river basins and their tributaries (Fig. 9). Notable examples include the Sabine, Trinity, Navasota, Guadalupe, and San Antonio River basins, where forested riparian zones consistently displayed ET/P values exceeding 100% (Fig. 9).

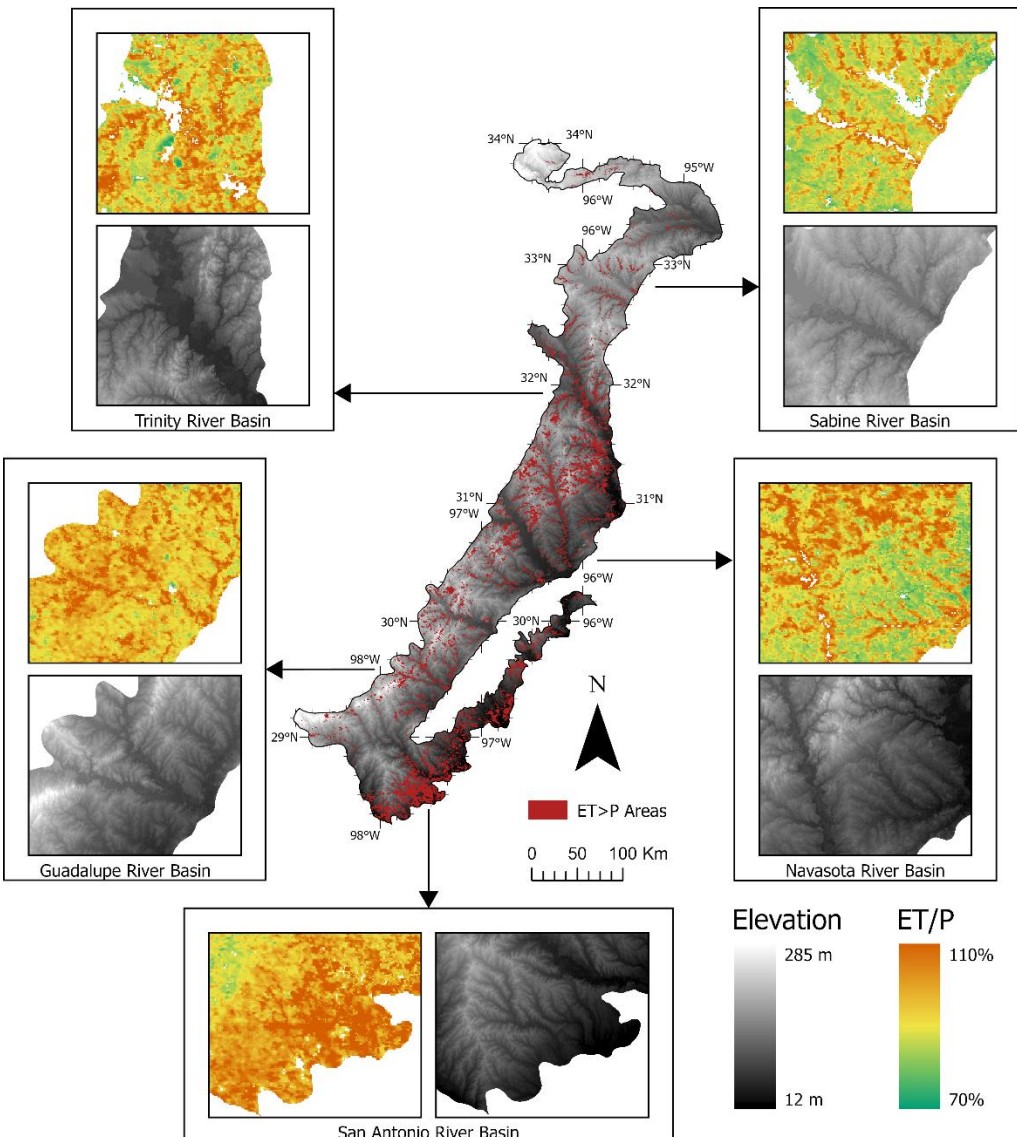

**Figure 9.** Spatial distribution of ET/P ratios and elevation across the Post Oak Savannah ecoregion. Areas with ET/P > 100% are highlighted in red, while grayscale shading indicates elevation. Insets show detailed views of selected river basins, illustrating the prevalence of high ET/P in forested, low-elevation regions (NASA, SRTM).

Total excess water varied substantially across the Post Oak Savannah ecoregion, with most values ranging from −5000 mm m$^{-2}$ to 7500 mm/m² (Fig. 10A). Over the entire study period (2009–2023), the mean excess water was 2422 mm m$^{-2}$, or 161 mm m$^{-2}$ per year (Fig. 10B). Temporally, excess water ranged from a low of −22,394,455 mm in 2022 to a high of 167,853,812 mm



in 2016 (Fig. 10C). On average, the ecoregion totalled 47,971,635 mm of excess water per year. Only two years exhibited negative excess water: 2011 (−21,968,413 mm) and 2022 (−22,394,455 mm) (Fig. 10C).

These contrasting totals reflect different hydrometeorological conditions. In 2011, precipitation was relatively low at 137 953 517 mm, whereas in 2022 it reached 221,613,882 mm (Fig. 10C). Conversely, evapotranspiration (ET) in 2011 was
also low, returning only 158,624,385 mm of water to the atmosphere compared with 242,295,663 mm in 2022 (Fig. 10C).

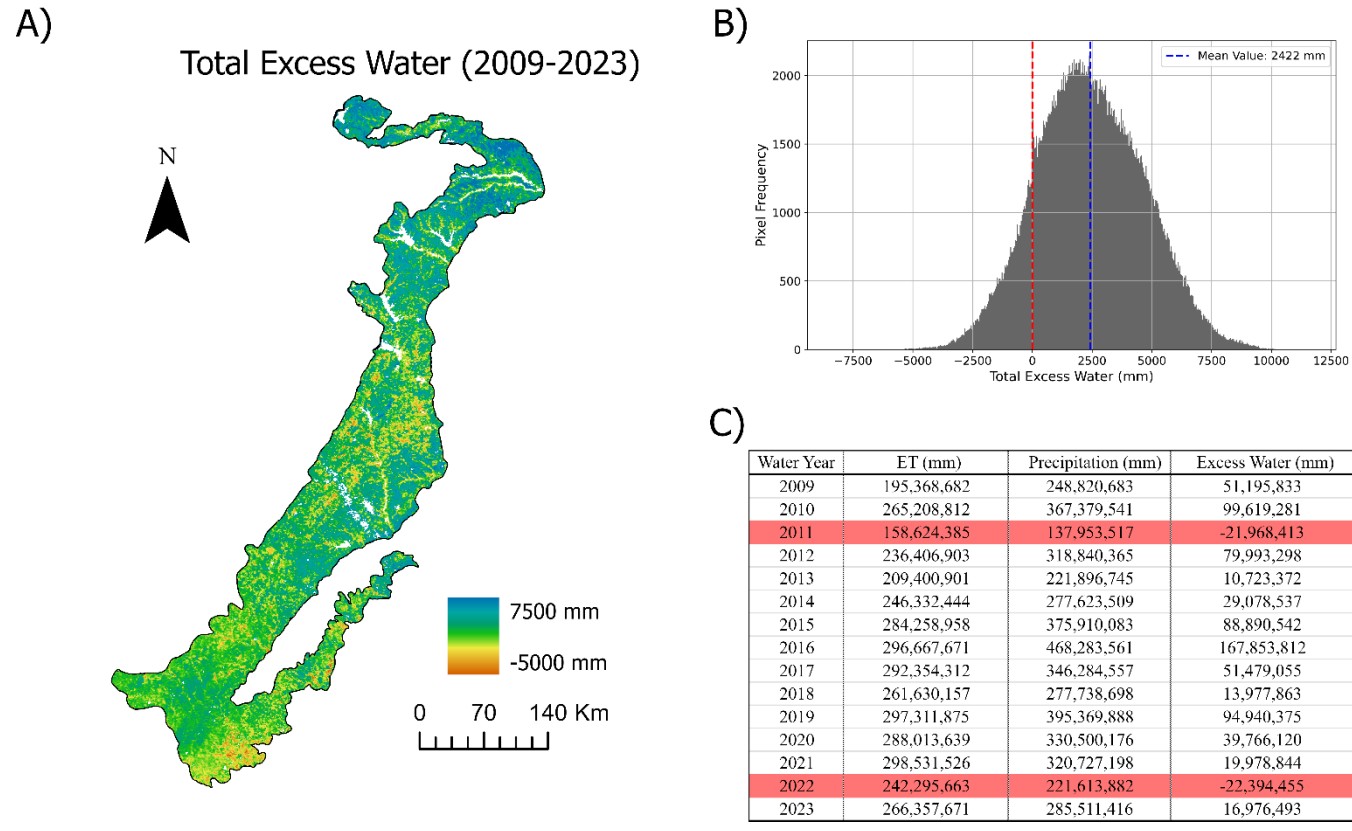

**Figure 10.** (A) Spatial distribution of total excess water (mm/m$^2$) across the Post Oak Savannah ecoregion from 2009 to 2023. The amounts range from −5000 mm/m$^2$ to 7500 mm/m$^2$. (B) Frequency distribution of total excess water across the ecoregion, with red and blue dashed lines indicating 0 total excess water and the mean, respectively. (C) Annual summary of ET, P, and resulting excess water (mm). Negative
values in the table (highlighted) indicate water years having net water deficits.

When integrating both woody vegetation cover and overall aridity into the analysis, increases in woody cover consistently reduced excess water totals across every precipitation zone (Fig. 11). The highest annual excess water (414.87 mm m$^{-2}$) occurs in areas with 0–10% woody cover that receive ≥1200 mm of precipitation (Fig. 11). In contrast, the lowest annual excess water (−122.87 mm m$^{-2}$) occurs in areas with ≥80% woody cover in the 600–800 mm precipitation zone (Fig. 11). Notably, none of
the 0–10%, 11–20%, 21–40%, or 41–60% woody cover classes exhibited negative excess water values (Fig. 11). Conversely, in the 61–80% and ≥80% woody cover categories, all precipitation zones had negative values except the ≥1200 mm zone, which remained positive (Fig. 11).




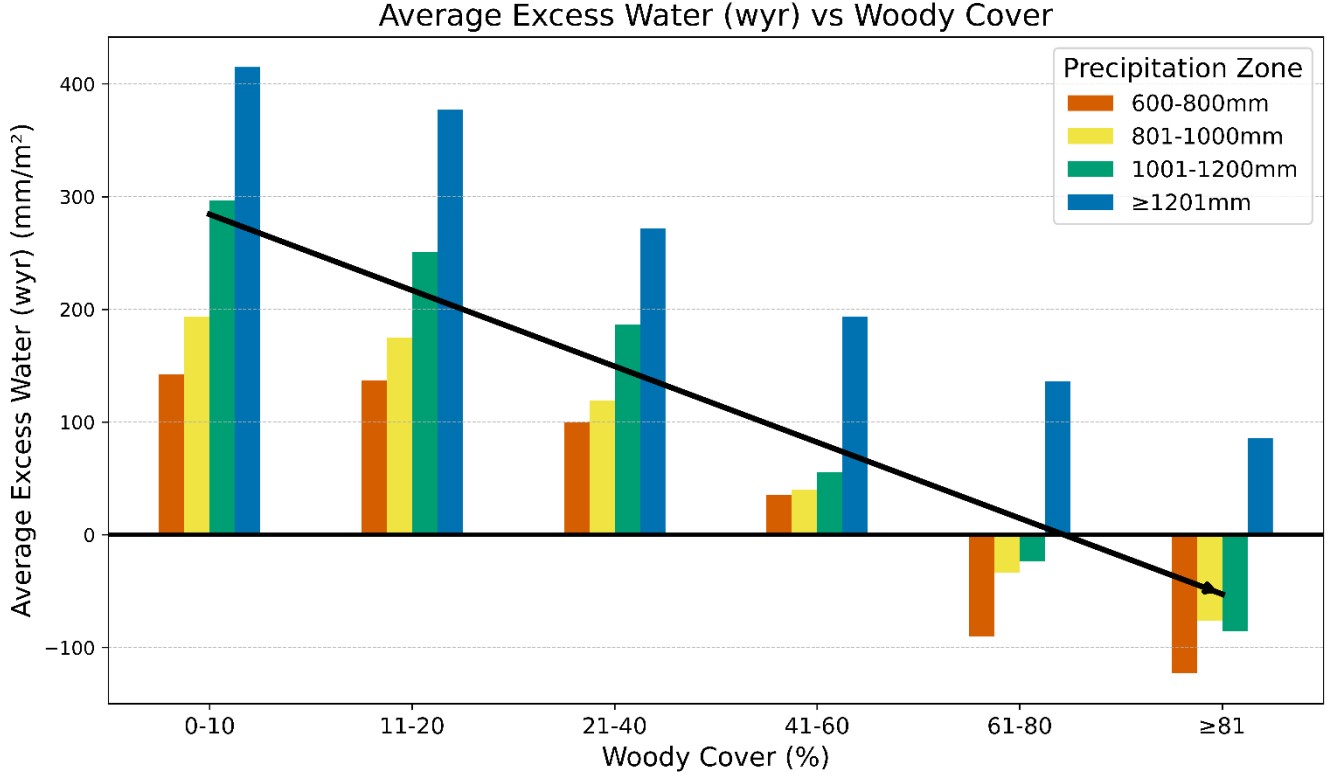

**Figure 11.** Mean annual excess water (mm/m²) as a function of woody cover (%) and precipitation zone. The black trend line indicates the overall decrease in excess water with rising woody cover. Positive values represent net surpluses, whereas negative values denote net deficits.

## 4 Discussion

### 4.1 MOD16 Accuracy

The MOD16 ET model used in this study showed moderate-to-strong agreement with WBET measurements at the HUC8 scale, with an overall RMSE of 103 mm $w^{-1}yr$ and an $R^2$ of 0.65 (Figure 4-4). The bias of −7 mm $w^{-1}yr$ indicates that the model generally neither overestimates nor underestimates ET. These findings align with other validations of the MOD16 ET product, performing better in some cases and worse in others (Aguilar et al., 2018; Nadzri & Hashim, 2014; Du & Song, 2018; Ruhoff et al., 2013; Miranda et al., 2017).

Miranda et al. (2017) reported higher accuracy (RMSE = 4.91 mm/month, $R^2$ = 0.82) in Brazil's Caatinga region, where annual temperature (±20 °C) and precipitation (300–1500 mm) are similar to those of the Post Oak Savannah. That monthly RMSE corresponds to an annual total of approximately 59 mm, outperforming the results of our study. Conversely, Ruhoff et al. (2013) observed that MOD16 overestimated ET by 19% compared with eddy covariance measurements in a Brazilian savannah, whereas our results indicate a slight tendency toward underestimation. Their $R^2$ (0.78) exceeded ours, but their RMSE (167.9 mm) was larger.





Comparisons with a semiarid region in Northwest Mexico that closely resembles the Post Oak Savannah in both
climate and species composition produced R² values ranging from 0.46 to 0.86, RMSE values from 142 to 168 mm, and biases
from −66 to −15 mm annually (Aguilar et al., 2018). These ranges are similar to our own and thus bolster the credibility of our
findings.

Interestingly, the subhumid northeastern HUC8s (HUC8s 1–4) exhibited the least accurate results (R² = 0.11–0.39,
RMSE = 92–104 mm), whereas those for semiarid central and southwestern HUC8s (HUC8s 7–11) were more accurate
(R² = 0.54–0.70, RMSE = 63–79 mm) (Figure 4-4). This outcome contrasts with the common pattern in MOD16 validation
studies, in which wetter climates or seasons typically yield better performance (Du & Song, 2018; Velpuri et al., 2013).

Overall, MOD16 showed variable results by tending to normalize extremes. For extremely dry years (i.e., 2011 and
2022), the model overestimated ET by 77 mm and 117 mm, respectively, suggesting inadequate constraints on ET under
conditions of low soil moisture or reduced stomatal conductance (Figure 4-4). In contrast, in wetter years (e.g., 2015 and 2016),
biases were lower (−62 mm and −80 mm, respectively), implying that increased cloud cover or unusual local conditions may
produce higher levels of ET than MOD16 accounts for (Figure 4-4).

Finally, 2009 was an outlier in terms of accuracy, likely influenced by drought followed by the effects of Hurricane
Ike, which made landfall on September 12, 2008, and impacted parts of the southern and eastern Post Oak Savannah. Rapid
vegetation changes, reflected in LAI and FPAR inputs, along with waterlogged soils, may have violated the model's
assumptions, leading to a miscalculation of the partitioning between evaporation and transpiration.

## 4.2 Monthly and Seasonal Trends

We observed peak precipitation in May and October, which aligns with the expected wet (spring and fall) and dry (summer)
seasons characteristic of a humid subtropical climate (Figure 4-5). Correspondingly, each canopy cover class showed its
highest ET rates in June (Figure 4-5). This peak likely results from a combination of actively growing vegetation, abundant
soil moisture following increased May rainfall, and warm temperatures that raise the vapor pressure deficit (VPD) and therefore
the atmosphere's capacity to hold water vapor (Liu et al., 2017; Sun et al., 2023). In addition, because our study area lies in the
northern hemisphere, the summer solstice occurs in June, providing heightening solar radiation, which further increases PET
(Aschonitis et al., 2017).

Despite July and August being the warmest months, ET declines substantially during this period (Figure 4-5). We
attribute this decrease to reduced soil moisture storage, which is rapidly depleted via high-VPD-induced transpiration and
evaporation (Mondal et al., 2024; Yang et al., 2023; Anav et al., 2018). This trend persists into the cooler months, when
temperatures begin to drop and rainfall increases, so that ET only begins to rise again only in February as temperatures rebound.

Interestingly, the ≥81 % canopy cover class does not exhibit the highest ET in every month, but only from April to
September (Figure 4-5). One explanation is that heavily "thicketized" (≥81%) woodlands may have a more complex vertical
structure— comprising both deciduous and evergreen species—such that the overlapping foliage layers produce a more closed
canopy during the warmer months (Whitehurst et al., 2013; Arumäe & Mait, 2018; Scott et al., 2015; Jucker et al., 2015). Many




thicketized Post Oak Savannah stands consist of oak overstory combined with an understory of evergreen species such as *Juniperus virginiana* and *Ilex vomitoria* (Olariu et al., 2024; Basant et al., 2023). In contrast, woodlands with 61%–80 % canopy cover are generally dominated by evergreen species (Pourrahmati et al., 2023; Arumäe & Mait, 2018; Stephens
et al., 2015), which remain active during cooler months—potentially explaining their higher ET from October through March. For instance, extensive *Pinus taeda* stands are found in the eastern Post Oak Savannah, adjacent to the Piney Woods ecoregion. Owing to their needle-shaped leaves, *Pinus taeda* woodlands typically range from 60 % to 80 % canopy cover, the higher percentages associated with mid-aged stands that include a mix of younger and older trees, eventually forming gaps in older stands (Song et al., 2009; Zeide & Stephens, 2010; Johnson et al., 2021).

Seasonal ET trends closely followed seasonal precipitation (Figure 4-6). From 2011 to 2014, monthly precipitation averaged 60 mm, resulting in minimal seasonal variation in ET. Between 2015 and 2021, however, monthly precipitation rose to an average of 86 mm—a 26 mm increase—which widened the seasonal stratification in ET. This increased stratification is attributable to higher transpiration rates during spring and summer, driven by the ample water supply that maintained elevated soil moisture (Fu et al., 2022; Koehler et al., 2023).

Notably, the severe drought of 2011 (Nielsen-Gammon, 2012; Chen et al., 2021) caused summer and fall ET to drop to winter-like levels (Figure 4-6). Although spring ET remained near average, this was likely a residual effect of the relatively wet conditions in 2009 and the average precipitation in 2010. The 2011 drought caused an estimated mortality of 65.6 (±7.3) million trees in East Texas alone—encompassing common Post Oak Savannah species such as *Quercus stellata*, *Quercus falcata*, *Ulmus alata*, and *Pinus taeda* (Klockow et al., 2018). Additionally, the difference between precipitation and
PET in 2011 reached −1206-mm (Schwantes et al., 2017). Widespread wildfires consumed nearly four million acres across Texas—31,453 individual fires—representing 47.3 % of all acreage burned by wildfire in the United States that year (Nielsen-Gammon, 2012; Texas A&M Forest Service, 2011).

### 4.3 Bioclimatic–ET Coupling

Evapotranspiration in the Post Oak Savannah showed a moderate positive relationship with precipitation and a weak negative
relationship with temperature (Figure 4-7). This P-ET coupling is consistent with global research findings, which highlights the tight linkage between these two fluxes across diverse ecosystems (Mondal & Mishra, 2024; Mondal et al., 2024; Xi et al., 2023; Zeng et al., 2010). Notably, the correlation was stronger in the more arid regions of the Post Oak Savannah (600–1000 mm vs. ≥1001 mm), where limited water availability acts as the primary constraint instead of energy inputs (e.g., radiation and temperature) (Nagler et al., 2007; Yu et al., 2021). Consequently, in these drier areas, ET begins soon after precipitation
events: soils rapidly absorb incoming rainfall, vegetation responds by increasing transpiration, and overall ET rises (Nielsen et al., 2024).

       In contrast, the negative relationship between ET and temperature may appear counterintuitive. However, many plants operate within an optimal temperature window for photosynthesis and transpiration (commonly 20°C –30°C) (Yamasaki et al., 2002; McGowan et al., 2020; Crous et al., 2022). In the Post Oak Savannah, severe summer heat and lower precipitation





often drive plants to close their stomata, thereby reducing transpiration despite high VPD. This negative relationship is
particularly strong in the most arid (600–800 mm) and most humid (≥1200 mm) areas, whereas it is weaker in the intermediate
(801–1200 mm) zone. In the arid region, limited soil moisture readily explains stomatal closure and reduced transpiration. In
more humid areas, factors such as persistent cloud cover or higher relative humidity may restrict the vertical movement of
water vapor from plant surfaces to the atmosphere (Wang et al., 2023; Dai et al., 1999).

Both canopy cover and canopy height exhibited positive relationships with ET, yet canopy height correlated more
strongly (Figure 4-7). Height provides a more integrative measure of forest water use by reflecting total aboveground biomass,
leaf area index (LAI), and vertical leaf stratification—all of which strongly influence transpiration and hence total ET
(Bonan, 2008; Baldocchi, 2003). These relationships remained relatively stable across all precipitation zones. Taller trees
typically develop deeper, more extensive root systems that enable access to subsurface water reservoirs, a vital adaptation

during the droughts often experienced in arid parts of the Post Oak Savannah. Consequently, such trees maintain transpiration
and growth even when upper soil layers are dry. For example, Horton and Hart (1998) describe hydraulic lift, whereby deep-
rooted trees transfer water from moist lower soil layers to drier surface layers, thereby enhancing moisture availability for
transpiration. Furthermore, Jackson et al. (2000) review the hydraulic architecture of trees and emphasize that taller individuals
often possess complex, far-reaching root systems. These systems improve the capacity to extract and transport water from

deeper sources, thus supporting a dense canopy and elevated transpiration rates.

### 4.4 ET/P and Excess Water

Over the study period, the average ET/P ratio in the Post Oak Savannah was 90% (Figure 4-8). Globally, the mean ET/P ratio
over land surfaces is approximately 65%, varying by continent. For instance, North America averages around 70%, whereas
Australia—which more closely resembles the Post Oak Savannah's overall conditions—exhibits a higher ratio of 87% (Reitz

et al., 2017; McDonald, 1961). An ET/P ratio of 90% is therefore plausible when compared with other semiarid or arid regions,
where studies have reported ratios between 80% and 93% (Fleischmann et al., 2023; Irmak, 2017). Moreover, Althoff and
Destouni (2023) suggest that ET/P will continue to rise as agricultural and forestry activities expand, increasing the prevalence
of trees—a pattern already observed in the Post Oak Savannah (Olariu et al., 2024). This high ratio also indicates an ET-driven
system, with evapotranspiration as the dominant water-budget component (Condon et al., 2020; Reitz et al., 2017), confirming

the assumption made by Basant et al. (2023).

The ET/P ratios above 100% observed in 2011 and 2022 likely resulted from severe drought conditions, forcing the
ecosystem to rely on minimal soil moisture reserves and possibly groundwater in riparian areas. Fleischmann et al. (2023)
reported similar findings in South American riparian zones, consistent with our Post Oak Savannah observations (Figure 4-9).

Excess water (P−ET) exhibited substantial spatial variability, ranging from −5000 mm m⁻² to over 7500 mm m⁻², with

an average of ~2500 mm m⁻² across the entire study period (Figure 4-10). However, 2011 and 2022 both showed net negative
excess water, explained by the same conditions that led to ET/P ratios exceeding 100%.



Of note, Figure 4-11 illustrates a steep decline in excess water with increasing woody cover. As tree and shrub cover expands, transpiration intensifies, lowering the net water surplus. These findings align with those of Basant et al. (2023), who found that understory shrub thicketization in the Post Oak Savannah substantially reduces groundwater recharge. Consequently, under continued WPE, the Post Oak Savannah will likely experience greater reductions in excess water—especially in its more arid regions, where soil moisture is already limited. Such changes may alter local water availability, affect aquifer recharge, and shift ecosystem functioning, as woody plants increasingly outcompete herbaceous vegetation for scarce moisture.

## 5 Conclusion

This study demonstrates that ET in the Post Oak Savannah is intricately linked to both climatic drivers and vegetation structure. Our analysis revealed a moderate positive relationship between precipitation and ET, confirming that water availability is a primary driver in this region. Conversely, temperature exhibited a weak negative relationship with ET—a finding that, while initially counterintuitive, can be explained by plant physiological responses such as stomatal closure during periods of extreme heat. In the context of global warming, rising temperatures coupled with increasingly sporadic precipitation are likely to exacerbate these dynamics. Higher temperatures not only elevate the atmospheric demand for water but also promote rapid soil moisture depletion, leading to more pronounced instances of water stress. This decoupling of energy and water fluxes ultimately underlines the importance of understanding the nuanced interplay between climate and hydrology in sustaining regional water resources.

Quantifying how variations in canopy cover affect water use and, thereby, regional hydrological processes is vital for evaluating the impacts of WPE and thicketization affect sustainable water management. Our findings indicate that as woody cover increases, excess water decreases—especially in arid regions—owing to enhanced transpiration. This reduction in net water surplus has significant implications for groundwater recharge and ecosystem functioning, as increasing woody vegetation competes with herbaceous species for limited moisture. Consequently, these shifts in vegetation structure demand adaptive management strategies to preserve water availability under future climate scenarios.

While our study employs robust remote sensing and hydrological modeling techniques, several limitations must be acknowledged. First, the absence of eddy covariance towers precludes direct, in-situ validation of the MOD16 ET product. However, validation at the HUC8 scale via water-balance estimates remains acceptable for a large-scale analysis. Second, the coarser spatial resolution of MOD16 (500 m) may mask fine-scale hydrological processes, particularly in irrigated agricultural areas. Future research employing higher-resolution ET datasets could improve the accuracy of these assessments. Lastly, although the 15-year study period captures critical periods of drought and high rainfall, even longer-term observations would further enhance our understanding of how continued global warming influences the interplay between temperature, precipitation, and ET.

Overall, these results provide a critical foundation for understanding how climatic changes and woody vegetation dynamics jointly shape regional water cycles. By quantifying the effects of canopy cover on ET and excess water across





different precipitation zones, this study informs land managers and policymakers facing the challenges of sustaining water

resources under ongoing global warming and WPE—not only in the Post Oak Savannah but in similar ecosystems worldwide.

## 6 Code availability

Woody Coverage code: https://code.earthengine.google.com/08f4a2fdce7672cb261f48fc658850e2

Sub-basin ET and P code: https://code.earthengine.google.com/c77b2aeb8fc4687677b33c1c141d16bc

ET/P and Excess water analysis code: https://code.earthengine.google.com/80ef181f4002d7314a10ae391800189d

Water Year aggregation code: https://code.earthengine.google.com/8b4ee77f99b3e067bae38c8386e150ff

Pointwise Sampling code: https://code.earthengine.google.com/1957d01209128479a368e655b5b75064

Monthly MODIS ET code: https://code.earthengine.google.com/2c21005c469551d5646b1ee86812cfe9

Monthly P and T code: https://code.earthengine.google.com/23bc61414ed99bb58892ea682a965b5e

## 7 Data availability

MODIS ET product: https://lpdaac.usgs.gov/products/mod16a2gfv061/

Daymet V4 Temperature product: https://daac.ornl.gov/cgi-bin/dataset_lister.pl?p=32

Canopy Cover product: https://rangelands.app/rap/?biomass_t=herbaceous&ll=36.5526,-101.3460&z=4&landcover_t=tre

Canopy Height products: https://lasers.tamu.edu/ice-cloudand-land-elevation-satellite-icesat-2-applications/ (Malambo and Popescu, 2024) https://glad.umd.edu/dataset/gedi (Potapov et al., 2021)

Runoff products: https://waterwatch.usgs.gov/index.php?id=romap3&sid=w__download

## 8 Author Contribution

Horia G. Olariu and Bradford P. Wilcox conceptualized the research goals and aims. Horia G. Olariu curated the data, performed the formal analysis, conducted the investigation, designed the methodology, prepared the data visualizations, and drafted the initial manuscript. Bradford P. Wilcox secured the project funding. Sorin C. Popescu and Bradford P. Wilcox

supervised the project and contributed to manuscript review.

## 9 Competing Interests

The authors declare that they have no conflict of interest



## 10 Acknowledgements

This research was supported by funding from the USDA National Institute of Food and Agriculture (NIFA). The authors also
gratefully acknowledge the infrastructure and resources provided by Texas A&M University, as well as the advanced remote
sensing facilities offered by the LASERS lab, which were instrumental in conducting the data analysis and ensuring the success
of this study.

## 11 Financial Support

This research was supported by the National Institute of Food and Agriculture (NIFA) under the project "THICKETIZATION
OF OAK SAVANNAS: CAN RESTORATION LEAD TO GREATER REGIONAL GROUNDWATER RECHARGE?"
(Accession No. 1027794, Grant No. 2022-67019-36267, Proposal No. 2021-09129)

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
