# Peer review of "Linking Woody Plants, Climate, and Evapotranspiration in a Temperate Savanna"

_EGUsphere, 2025_

## Author Response (AR1)

**Response Letter**

**\*Reviewer comment in bold black font**

\*Author Response in red font

**Reviewer 1**

We thank the reviewer for their helpful comments, which have undoubtedly strengthened and refined the manuscript. Below, we respond to each of their specific points and explain how we have addressed them.

**Q1. In Section 3.1, the comparison of the MOD16 product with WBET estimates showed good overall accuracy, but performance was very low between 2009–2011. Even 2018 and 2022 can be considered as years of poor performance, since the R² did not reach 0.5. In addition, sub-basins 1–3 showed low accuracy. It might be better not to include these years and sub-basins in the subsequent analyses, as the evapotranspiration estimates are not reliable and could introduce bias into the interpretation of results.**

**Moreover, a more detailed explanation should be provided in the discussion (Section 4.1) about why satellite estimates performed poorly in these years and sub-basins. You mention the effects of Hurricane Ike and that performance is worse in dry years (2011 and 2022), but 2009 and 2010 also show low performance despite precipitation being closer to the average. The MOD16 product performs better in drier regions than in wetter ones. Therefore, why does accuracy decrease in dry years if the product tends to perform better in dry conditions? It would be helpful to elaborate on why performance was poor in those years as well. Additionally, although you mention that performance is lower in HUC8s 1–4, possible reasons are not discussed.**

Thank you for highlighting the low-accuracy years (2009–2011, 2018, 2022) and the weaker performance in sub-basins 1–3. We have retained all years and sub-basins because our study focuses on basin-wide, multi-year relationships between evapotranspiration (ET) and its bioclimatic drivers; those relationships are evaluated within precipitation zones and through newly added non-linear GAMs that lessen the influence of any one year or sub-basin. Removing the low-performance cases would also eliminate the very climatic extremes and spatial heterogeneity that reveal the strengths and limitations of remote-sensing ET products and are central to our uncertainty analysis.

To clarify why accuracy varied, we have rewritten Section 4.1. We added L352–354—
**"Performance varied markedly among years. In 2009–2010, annual rainfall totals were near the long-term mean, but precipitation was concentrated in northern catchments and deficient in the south, creating north–south gradients that the basin-wide WBET captured but MOD16 ET did not, thereby showing increased RMSE values."** We then inserted L354–362— **"During the extreme droughts of 2011 and 2022, MOD16 over-estimated ET by 77 mm and 117 mm (Fig. 4), respectively, exposing a known weakness in the algorithm in representing soil-moisture stress when stomatal conductance is modelled from meteorology**

**Response Letter**

alone (Hu et al., 2015; Miralles et al., 2016; Majozi et al., 2017). Therefore, although our semi-arid basins showed comparatively higher accuracy overall, the literature shows that MOD16 often struggles in arid and semi-arid environments because it lacks an explicit soil-moisture constraint—making overprediction likely when soils are critically dry even within otherwise well-performing regions. By contrast, biases in wetter years—such as 2015 and 2016 (−62 mm and −80 mm, respectively; Fig. 4)—were modest but still larger than those in average-precipitation years. These residual errors may reflect reduced available energy under persistent cloud cover and/or enhanced flood-plain evaporation that raised actual ET beyond what MOD16 captured." Finally, we concluded with L373–375— "**MOD16 normalizes extremes—over-estimating ET when soils are parched and under-estimating it in complex, water-rich mosaics—highlighting the need for soil-moisture constraints in future versions.**" These additions preserve the original points while providing clearer, citation-supported reasoning for the performance patterns noted by the reviewer.

**Q2. Consider displaying Figure 7 as a 2 × 2 panel to increase the size of the scatterplots.**

We agree and reconstituted Figure 7 as a 2x2 panel.

**Q3. In the discussion section, all figures are referenced as "Figure 4" (e.g., Figure 4–5, Figure 4–6, etc.). I assume this is a mistake, as Figure 4 is only relevant to the accuracy of the validation.**

This was mistake on our end, which we have rectified. Please notify us if we have missed revising any figure references.

**Q4. In Section 4.3, you explain that there is a negative relationship between temperature and ET, and that the landscape includes a mix of deciduous and evergreen vegetation. Usually, evergreen vegetation can reduce their transpiration in summer (water saver) but deciduous vegetation increases it due to higher water demand (water spender). Therefore, under higher temperatures, ET would be expected to increase in deciduous vegetation. You might consider better explaining the differences between vegetation types (evergreen vs. deciduous) across the region and their role in ET.**

**Also, the relationship between temperature and ET is usually non-linear. Higher temperatures increase ET up to a threshold, after which ET decreases due to stomatal closure (as you explain in the section). It might be useful to include a non-linear analysis, such as a Generalized Additive Model (GAM), to test whether there is a positive relationship up to a certain threshold. Therefore, temperature does not have a strictly negative effect on ET, as its impact depends on the temperature range.**

To address this comment, we included a non-linear (GAM) analysis for each independent bioclimatic variable (including air temperature) which you can see in the revised Figure 7. Furthermore, we added L432-440 "**GAM fits (magenta curves in Fig. 7) revealed a non-linear,**

**Response Letter**

dome-shaped response of annual ET to mean air temperature. ET climbed steadily to a peak at ≈ 22–24 °C, plateaued, and then declined above ~25 °C; the GAM pseudo-R² was 0.13, only marginally higher than the aggregated linear R² (0.11), but it captured the threshold beyond which stomatal regulation suppresses transpiration. This pattern is consistent with the divergent thermal strategies of the region's dominant woody species. The evergreen loblolly pine (*Pinus taeda*) begins to reduce stomatal conductance at leaf temperatures near 32 °C, whereas drought-deciduous post-oak (*Quercus stellata*) and blackjack oak (*Q. marilandica*) maintain higher conductance until ≈35 °C before closing their stomata (Oren et al., 1999; Novick et al., 2016). Because summer days in the Post Oak Savannah frequently exceed these thresholds, particularly during drought years, elevated mean annual temperatures integrate numerous midday periods of stomatal closure, driving down yearly ET despite higher vapor-pressure deficits." Which should provide context for the dominant vegetation types (evergreen vs deciduous) across the Post Oak Savannah.

**Reviewer 2**

We thank the reviewer for their positive remarks and for the fair, constructive criticism that has certainly strengthened our manuscript. Below, we respond to each of their specific points and explain how we have addressed them.

**Introduction**

**L27: I suggest to use another acronym for temperature. T is often used to refer to transpiration in the ET modeling community and I also suggest to be more specific throughout the text and refer to air temperature (Ta) rather than just temperature, which could be confused with land surface temperature (LST).**

We agree that "T" can denote several variables, including transpiration and land-surface temperature. Accordingly, we reviewed the manuscript and replaced every instance of temperature (T) with air temperature (Ta), including in Figures 1, 2, 7, and 8.

**L38-39: '[…] observed ET decreases of 31.9 mm and 110 mm, respectively'. I assumed this is at annual scale? If so, add mm/year.**

Correct. We added mm yr$^{-1}$ to indicate the annual scale.

**L71: I don't think it is correct to use the term 'validate' when you only compare the MOD16 product with water balance method. This is more of a comparison rather than any kind of validation since no observed benchmark values are used, since the water balance method is subjected to uncertainties in the precipitation/runoff products and assumptions made about other processes at annual scales (groundwater recharge, storage etc).**

**Response Letter**

You are correct that, because the WBET method carries its own uncertainties and assumptions, our analysis should not be described as a "validation." Accordingly, we have replaced every instance of "validation" with "evaluation" throughout the manuscript.

**Materials and methods**

**Figure 1. Please add the data sources for each of variables (precipitation, air temperature and canopy cover). Although it is specified in section 2.2, figure captions should be interpretable as much as possible without refereeing to the text. Also please add details about the air temperature as similarly done with mean annual precipitation. It is annual daytime average? Also mean annual precipitation is calculated using which years?**

We rewrote the caption to reflect your useful recommendations. It now reads "**Overview of the Post Oak Savannah ecoregion in east-central Texas. Panel (A) places the ecoregion within the conterminous United States, highlights Texas and the Carrizo–Wilcox Aquifer, and overlays a 2023 true-color Landsat 8 OLI mosaic. Panels (B) and (C) draw on the 2008–2023 Daymet V4 daily precipitation record: panel (B) maps mean annual precipitation (MAP, mm yr⁻¹), calculated as the multi-year average of the annual sums of daily totals, and panel (C) reclassifies that MAP surface into four precipitation zones (600–800, 801–1000, 1001–1200, and ≥1201 mm). Panel (D) depicts mean annual air temperature (MAT, °C) for the same period, derived from Daymet V4 by averaging daily maximum and minimum air temperatures [(T$_{max}$+T$_{min}$)/2] and then averaging those daily means across 2008–2023. Panel € presents fractional canopy cover (%) at 30 m resolution from the Rangeland Analysis Platform V4, averaged over the identical 2008–2023 window. Specifying these data sources, periods, and processing steps allows the caption to be interpreted independently of the main text.**"

**2.3.1 MOD16 ET validation: Again, please consider changing the sub-title since a validation is not actually done. At most, it can be considered an 'evaluation' or 'benchmarking'.**

We changed the subtitle to "**MOD16 ET Evaluation**"

**Figure 3. Add units in table in the Total Area column.**

We added units to the Total area column - **(km$^2$)**

**Results**

**Figure 7. I suggest to also add the R2 aggregating for all precipitation regimes along with separating them for each (600-800, 800-1000, 1000-1200, >1200nm) as the authors did. This might better depict the general tendencies and contrast better if different precipitation regimes (or eco-regions) show different relationships with each of the variables assessed.**

**Response Letter**

**For example, the relationship between canopy height and ET has very similar slopes and R2 for all precipitation regimes, while the other variables show large differences.**

We agree with this point and added in bold the aggregated slope and $R^2$ for each of the variables assessed.

**L294: '[…] when ET/P exceeded 100%'. This directly contradicts the earlier statement when the authors say that ET/P ranged between 70 and 100%.**

After rereading this section, we saw that it needed clearer wording. We have revised it to read (L303–305): **"The ratio of ET to P (ET/P) remained fairly stable over the study period, averaging 90 % and ranging from 70 % to 100 % in most years (Fig. 8). The only exceptions were the drought years 2011 and 2022, when ET/P rose slightly above 100 %. Both years were marked by above-average air temperatures and below-average precipitation (Fig. 8)."**

**Discussion**

**4.1 MOD16 accuracy: I suggest to mention and discuss the possible uncertainties and limitations of using the water balance method as a benchmark in this section.**

This omission was an oversight, and the original manuscript should have acknowledged the uncertainties in the WBET evaluation. We have now included a new passage (lines 346–351) that highlights four key sources of error: **"(1) gauge-based precipitation grids may be biased by undercatch and sparse station coverage; (2) several sub-basins extend beyond the Post Oak Savannah boundary, so lateral inflows and outflows can distort basin averages; (3) long-term soil- and groundwater-storage changes are assumed negligible even though seasonal drought–recharge cycles can shift storage by several centimetres; and (4) small reservoirs and irrigation withdrawals remain in the streamflow record, potentially inflating inferred ET during dry years."**

**4.1 MOD16 accuracy: many studies have shown that MOD16 does not perform well in arid /semi-arid ecosystems, mostly since the model does not properly capture plant water stress, especially stress related to soil moisture deficit since the MOD16 product models stomatal conductance solely based on meteorological data. Here are some studies, in case it could be relevant to contextualize better the MOD16 evaluation done in this study:**

The reviewer correctly points out that MOD16's underperformance in arid and semi-arid ecosystems is well documented, despite multiple updates and new versions. We have therefore added the supporting studies the reviewer provided to our manuscript. Lines 355–360 now read: **"During the extreme droughts of 2011 and 2022, MOD16 overestimated ET by 77 mm and 117 mm, respectively (Fig. 4), exposing a known weakness in the algorithm: soil-moisture stress is represented only indirectly when stomatal conductance is modelled from meteorology alone (Hu et al., 2015; Miralles et al., 2016; Majozi et al., 2017). Thus,**

**Response Letter**

**although our semi-arid basins performed comparatively better overall, the literature shows that MOD16 often struggles in arid and semi-arid environments because it lacks an explicit soil-moisture constraint—making overprediction likely when soils are critically dry, even in otherwise well-performing regions.**"

**L375-376: How come the authors didn't relate LAI with ET? This may better capture phenological differences and is more related to how much radiation is intercepted to transpire/photosynthesis than canopy cover.**

Thank you for the suggestion. We selected canopy cover instead of LAI because the RAP fractional-cover product offers higher resolution and fewer data gaps than the available MODIS LAI in our study area. Canopy cover is also the metric most familiar to local land managers facing rapid woody plant encroachment, so using it keeps the results practical and easy to communicate. Finally, canopy cover complements canopy height by adding a horizontal dimension to the vertical information already analyzed.

**L425: canopy height is also an important indicator of surface roughness which can influence the aerodynamic resistance to water transport from surface to atmosphere. Higher canopy height may enhance turbulent conditions and promote transpiration.**

This is an excellent point. We have incorporated it into the manuscript (L447–449): "**Taller canopy structures increase surface roughness, which lowers aerodynamic resistance and enhances turbulent exchange, thereby promoting more efficient transfer of water vapor from the canopy to the atmosphere.**"

**L436: I suspect the ET/P ratios above 100% may be also due to model uncertainties in the MOD16 product. Indeed, as mentioned in previous comment, MOD16 does not capture very well plant water stress, which likely would have been very high in those severe drought years leading to an overestimated ET and, potentially, higher values than P.**

We agree with this assessment and incorporated it into the manuscript (L466–468): "**Ratios above 100% may also stem from uncertainties in the MOD16 product, which does not adequately represent plant water stress; under the extreme stress of 2011 and 2022, MOD 16 likely overestimated ET, producing values higher than precipitation.**"

**L465: the authors mention that the future direction should be to use higher spatial resolution ET products to capture fine hydrological processes but I would rather suggest to explore other ET products particularly those based on Land Surface Temperature (LST) from thermal infrared (TIR) remote sensing, which have been shown to better capture plant water stress which is an important issue in water limited savanna ecosystems. See these studies:**

Thank you for pointing out that we had not discussed surface energy-balance algorithms that leverage thermal-infrared (TIR) remote sensing. We have corrected this oversight by adding the

**Response Letter**

following sentence to the manuscript (lines 468–501): "**Future work should also explore ET products that integrate thermal-infrared land-surface temperature data—such as Sentinel-2/3 fusion approaches or two-/three-source energy-balance models—which can better diagnose plant-water stress in semi-arid savannas (Guzinski et al., 2020; González-Dugo et al., 2021; Burchard-Levine et al., 2022; Anderson et al., 2024).**"